# Applying K-Means Cluster Analysis to Urinary Biomarkers in Interstitial Cystitis/Bladder Pain Syndrome: A New Perspective on Disease Classification

**DOI:** 10.3390/ijms26083712

**Published:** 2025-04-14

**Authors:** Yuan-Hong Jiang, Jia-Fong Jhang, Jen-Hung Wang, Ya-Hui Wu, Hann-Chorng Kuo

**Affiliations:** 1Department of Urology, Hualien Tzu Chi Hospital, Buddhist Tzu Chi Medical Foundation, Hualien 970, Taiwan; redeemerhd@gmail.com (Y.-H.J.); alur1984@hotmail.com (J.-F.J.); ryoma499@gmail.com (Y.-H.W.); 2Department of Urology, School of Medicine, Tzu Chi University, Hualien 970, Taiwan; 3Department of Medical Research, Hualien Tzu Chi Hospital, Buddhist Tzu Chi Medical Foundation, Hualien 970, Taiwan; jenhungwang2011@gmail.com

**Keywords:** K-means cluster analysis, urinary biomarkers, interstitial cystitis, oxidative stress, inflammation

## Abstract

This study applied K-means cluster analysis to urinary biomarker profiles in interstitial cystitis/bladder pain syndrome (IC/BPS) patients, aiming to provide a new perspective on disease classification and its clinical relevance. We retrospectively analyzed urine samples from 127 IC/BPS patients and 30 controls. The urinary levels of 10 inflammatory cytokines and three oxidative stress markers (8-hydroxy-2-deoxyguanosin [8-OHdG], 8-isoprostane, and total antioxidant capacity [TAC]) were quantified. K-means clustering was performed to identify biomarker-based patient subgroups. IC/BPS patients exhibited significantly elevated urinary levels of Eotaxin, MCP-1, NGF, 8-OHdG, 8-isoprostane, and TAC compared to controls (all *p* < 0.05). K-means clustering identified four distinct subgroups. Cluster 4, characterized by the highest levels of inflammatory and oxidative stress biomarkers, comprised 85% ESSIC type 2 IC/BPS patients and exhibited the lowest visual analogue scale (VAS) pain scores and maximal bladder capacity (MBC). Correlation analysis revealed distinct cluster-specific associations between biomarker levels and clinical parameters, including the VAS pain score, MBC, the grade of glomerulation, and treatment outcomes. Applying K-means clustering to urinary inflammatory and oxidative stress biomarkers provides a new perspective on disease classification, identifying IC/BPS subtypes with distinct clinical and biochemical characteristics. This approach may refine disease phenotyping and guide personalized treatment strategies in the future.

## 1. Introduction

Interstitial cystitis/bladder pain syndrome (IC/BPS) is a chronic inflammatory bladder disorder with a multifactorial etiology, including urothelial dysfunction, neurogenic inflammation, mast cell activation, autoimmunity, and occult infection [1,2]. Its clinical presentation is heterogeneous, encompassing various phenotypes that influence treatment outcomes [3]. These make the diagnosis and classification challenging in IC/BPS. Despite advancements in biomarker research [4,5], no single biomarker has been validated for the clinical use of IC/BPS, and existing classification methods, primarily based on symptoms or cystoscopic findings, may not fully capture the molecular heterogeneity of the disease.

A review has highlighted the role of oxidative stress in IC/BPS pathogenesis, with reactive oxygen species impairing bladder function through diverse molecular mechanisms [6]. Studies have demonstrated that IC/BPS patients exhibit distinct urinary oxidative stress biomarkers and inflammatory cytokine profiles compared to control groups [7,8]. Moreover, the levels of these biomarkers in urine were positively correlated with the grade of glomerulation and negatively correlated with maximal bladder capacity (MBC), although the correlations were weak. One retrospective study further revealed that the lower urinary levels of the regulated upon activation, normal T cell expressed, and presumably secreted (RANTES), 8-isoprostane, macrophage chemoattractant protein-1 (MCP-1), and 8-hydroxy-2-deoxyguanosine (8-OHdG) could predict satisfactory treatment outcomes in IC/BPS patients [9]. Despite these findings suggesting that urinary biomarkers have potential as diagnostic and prognostic tools for IC/BPS, the presented violin plots of biomarker levels demonstrate substantial overlap between IC/BPS patients and controls. Moreover, these studies primarily analyzed the role of individual biomarkers rather than providing a comprehensive assessment of the overall biomarker profile. These limitations may restrict the clinical applicability of urinary biomarker profiling in IC/BPS.

Given the heterogeneous nature of IC/BPS and the overlapping urinary biomarker profiles between patients and controls observed in previous studies, traditional biomarker analysis methods may not be sufficient for distinguishing clinically relevant subgroups. K-means clustering, an unsupervised machine learning technique, has been widely studied and applied in a variety of areas [10,11]. Integrating K-means clustering with urinary biomarker profiling might facilitate the identification of biologically meaningful IC/BPS subgroups within complex datasets, providing a more objective and data-driven classification approach.

This study applies K-means clustering to urinary inflammatory and oxidative stress biomarkers in IC/BPS patients to explore whether biomarker-based clustering can reveal clinically distinct disease subgroups. We hypothesize that this approach will provide a more objective and reproducible method for IC/BPS classification, offering insights into disease heterogeneity and treatment response.

## 2. Results

A total of 127 IC/BPS patients and 30 control subjects were included in the study (Table 1). The mean age of IC/BPS patients was 54.6 ± 12.6 years, while that of the control group was 58.9 ± 10.8 years (*p* = 0.083). The study cohort consisted of 127 IC/BPS patients who were predominantly female (89.8%), with 29.1% (n = 37) classified as European Society for the Study of Interstitial Cystitis (ESSIC) type 1 and 70.9% (n = 90) as type 2. The mean MBC was 711.4 ± 179.5 mL. The mean symptom severity scores of IC/BPS patients were as follows: Interstitial Cystitis Symptom Index (ICSI): 10.2 ± 4.5; Interstitial Cystitis Problem Index (ICPI): 10.2 ± 4.0; O’Leary–Sant symptom score (OSS): 20.6 ± 7.9; visual analogue scale (VAS) pain score: 4.2 ± 2.7. The grades of glomerulation were distributed as follows: 29.1% grade 0, 32.3% grade 1, 33.9% grade 2, 3.9% grade 3, and 0.8% grade 4.

Urinary biomarker analyses revealed significant differences between IC/BPS patients and controls (Table 2). Notably, the levels of Eotaxin, MCP-1, nerve growth factor (NGF), 8-OHdG, 8-isoprostane, and total antioxidant capacity (TAC) were significantly elevated in IC/BPS patients compared to controls (all *p* < 0.05).

K-means clustering grouped the 13 targeted urinary biomarkers into four distinct clusters among IC/BPS patients and controls (Figure 1). These four clusters had sizes of 53, 80, 4, and 20 patients. Cluster 3, consisting of only four cases (2.5%) (Case 18, Case 62, Case 115, and Case 134), exhibited highly elevated levels of IL-6, IL-8, chemokine (C-X-C motif) ligand 10 (CXCL10), macrophage inflammatory protein 1β (MIP-1β), and tumor necrosis factor α (TNFα). The hierarchical clustering of urinary biomarker profiles further supported the inference that Cluster 3 represents outliers (Supplement Appendix A).

### 2.1. Cluster Analysis and Clinical Characteristics

Table 3 summarizes the demographic and clinical characteristics of the study population categorized into four clusters based on the K-means clustering of urinary biomarker profiles. Sex distribution varied significantly (*p* = 0.047), with Cluster 4 having the highest proportion of male participants (25.0%). Cluster 2 included the majority of ESSIC type 1 IC/BPS patients (n = 23, 62.2%) and controls (n = 18, 60.0%), while Cluster 4 had a high proportion of ESSIC type 2 patients (85%, n = 17). VAS pain scores and MBC differed significantly among clusters, with Cluster 4 showing the lowest values (VAS: 2.80 ± 2.33, *p* = 0.033; MBC: 581.0 ± 183.70 mL, *p* = 0.003). Other clinical symptom scores and glomerulation grades showed no significant differences. Treatment outcomes were comparable across clusters; however, none of the patients in Cluster 4 achieved marked improvement (Global Response Assessment [GRA] = +3), which was significantly lower than in the other clusters (*p* = 0.036).

### 2.2. Urinary Biomarker Profiles in Different Clusters

Table 4 revealed the distinct urinary biomarker profiles across the four clusters. Cluster 4 had the highest levels of oxidative stress biomarkers (8-OHdG, 8-isoprostane, and TAC) and inflammatory biomarkers (Eotaxin, IL-6, CXCL10, MCP-1, and RANTES). In contrast, Cluster 2 exhibited the lowest levels of most biomarkers among all clusters. Cluster 1 showed intermediate biomarker levels, which were higher than Cluster 2 but lower than Cluster 4, including Eotaxin, IL-2, CXCL10, RANTES, and 8-OHdG.

### 2.3. Correlations Between Urinary Biomarkers and Clinical Parameters

Figure 2 presents the correlation coefficients between urinary cytokine levels and clinical characteristics within each cluster (excluding controls). Cluster 1 exhibited weak to moderate correlations between urinary biomarker levels and cystoscopic hydrodistention parameters, including a negative correlation between MBC and Eotaxin (r = −0.364) and positive correlations between glomerulation grade and IL-2 (r = +0.334) and 8-OHdG (r = +0.439). Cluster 2 exhibited weak correlations between urinary biomarkers, pain severity, cystoscopic hydrodistention parameters, and treatment response. TNFα correlated positively with VAS (r = +0.272) and negatively with GRA (r = −0.292), while CXCL10 (r = −0.256) and RANTES (r = −0.302) correlated negatively with MBC. 8-OHdG showed a weak positive correlation with glomerulation grade (r = +0.272). Cluster 4 displayed moderate to strong correlations between urinary biomarker levels and clinical symptoms, pain severity, and treatment response. 8-isoprostane correlated positively with ICPI (r = +0.573) and VAS (r= +0.468), while TAC correlated positively with VAS (r = +0.598). MIP-1β was negatively associated with GRA (r = −0.597).

A summary of each cluster’s key characteristics is presented in Table 5.

To further evaluate the clustering performance within confirmed IC/BPS patients, an additional cluster analysis excluding the control group was performed (Appendix A). This supplementary analysis yielded four distinct clusters, including one potential outlier group, and supports the robustness of the urinary biomarker-based clustering approach.

## 3. Discussion

The findings of this study underscore the heterogeneity of IC/BPS and highlight the potential of biomarker-based clustering as an objective approach to disease classification. By applying K-means clustering to urinary inflammatory and oxidative stress biomarkers, we identified four distinct patient subgroups. These biomarker-based subgroups exhibited different clinical characteristics and treatment responses, further supporting the notion that IC/BPS is a syndrome with multiple phenotypic subtypes. While the underlying pathophysiological mechanisms remain to be fully elucidated, the distinct urinary biomarker profiles observed across subgroups suggest potential biological heterogeneity within the syndrome.

Traditional IC/BPS classification is based on clinical symptoms and/or cystoscopic findings [12,13]. However, these methods may not capture the molecular complexity of the disease. Our study demonstrated that applying K-means clustering to urinary biomarkers provides deeper insights into IC/BPS patient subtypes, revealing significant variations in inflammatory and oxidative stress marker levels among clusters. Cluster 4, comprising 12.7% of the total cohort, predominantly consisted of ESSIC type 2 IC/BPS patients and exhibited the highest biomarker levels, suggesting a more inflammatory-driven disease process with increased oxidative stress. In contrast, Cluster 2, accounting for 51.0% of the total, included the majority of ESSIC type 1 IC/BPS patients and controls and displayed the lowest biomarker levels, reflecting a less severe bladder inflammatory disease subtype. Cluster 3, representing 2.5% of the total, was considered an outlier cluster in this new classification. This was consistent with the outlier percentage (less than 5%) reported in a previous study [7]. These findings support the heterogeneous nature of IC/BPS [3,14] and highlight the potential of urinary biomarker-based subtyping in refining disease classification.

A recent review discussed the potential role of urinary biomarkers in IC/BPS and their impact on therapeutic outcomes, but only to a limited extent [15]. Our results (Table 5) revealed varying cluster-specific correlations between urinary biomarkers and clinical parameters among different clusters, reinforcing the relevance of biomarker-based subtyping in predicting disease severity and treatment response.

Notably, Cluster 4 patients, characterized by elevated oxidative stress markers (8-OHdG, 8-isoprostane, and TAC) and inflammatory cytokines (Eotaxin, IL-6, CXCL10, MCP-1, and RANTES), exhibited lower MBC and lower VAS pain scores. Furthermore, 8-isoprostane levels showed a moderate to strong positive correlation with ICPI (r = +0.573) and VAS (r = +0.468), which was significantly stronger than the weak correlations observed in previous studies of the entire ESSIC type 2 IC/BPS cohort [7,8]. These findings suggest that Cluster 4 patients experience pronounced oxidative stress and more severe bladder inflammation, manifesting as reduced bladder capacities yet lower VAS pain scores. One possible explanation for this contradictory finding (i.e., lower MBC yet lower VAS) is that chronic or intense inflammation can lead to partial nerve desensitization, tissue remodeling or fibrosis, altered pain thresholds, or behavioral adaptations (e.g., frequent voiding that minimizes overdistension), thereby diminishing perceived pain. Consequently, despite more severe structural damage and elevated oxidative stress and inflammatory markers, such alterations in sensory signaling and pain modulation might attenuate pain perception. 8-isoprostane may serve as a potential biomarker for disease severity in this subgroup. Moreover, MIP-1β levels demonstrated a strong negative correlation with GRA (r = −0.597), suggesting its potential as a predictor of poor treatment response in Cluster 4 patients.

Although the treatment success rate (GRA ≧ +2) was similar across clusters, Cluster 4 exhibited a significantly lower marked improvement rate (GRA = +3) compared to Cluster 1 and Cluster 2 after treatment. This finding underscores the varying treatment responsiveness among biomarker-based clusters and highlights the importance of personalized treatment strategies. Patients with stronger inflammation (Cluster 4) may experience fewer benefits from conventional treatments and could require more aggressive or alternative therapeutic approaches. This finding is consistent with the previous study showing that treatment non-responders had higher urinary inflammatory cytokine levels than responders before treatment [16].

In our study, urinary NGF levels were significantly higher in IC/BPS patients compared to the controls (Table 2), which is consistent with previous findings [17,18]. However, no significant differences in NGF levels were observed across the identified clusters (Table 4). This suggests that while urinary NGF is useful for distinguishing IC/BPS patients from controls, it may not be sufficient for differentiating subtypes within the IC/BPS population. The clustering was based on the combined profile of 13 urinary biomarkers and not NGF alone. This also highlights the importance of a comprehensive assessment using the overall biomarker profile—such as through K-means cluster analysis—rather than relying on individual biomarkers in isolation.

K-means clustering has been used to identify symptom-based subtypes of patients with IC/BPS and chronic prostatitis/chronic pelvic pain syndrome [19]. The study found that three symptom-based subtypes exhibited distinct pelvic and systemic presentations; however, it also emphasized the need for further research to confirm whether these clusters differ in pathophysiology. In contrast, our study is the first to apply K-means clustering for biomarker-driven subgrouping in IC/BPS classification. By analyzing urinary biomarkers with K-means clustering, we identified distinct subgroups with unique clinical characteristics and treatment responses. This biomarker-based approach offers a more objective and reproducible method for disease stratification compared to traditional classifications based solely on clinical presentation.

Recently, machine learning models, including decision tree models, have been applied to IC/BPS biomarker research, demonstrating high accuracy in disease diagnosis (AUC = 0.87) [20] and treatment outcome prediction (81% accuracy) [9]. A comparative analysis of machine learning methodologies further highlighted the advancements in IC/BPS diagnosis, showing that auto-machine learning approaches outperformed traditional models, achieving an AUC of up to 0.96 through the integration of urinary biomarker data [21]. These findings underscore the potential of machine learning in precision medicine for IC/BPS, enabling biomarker identification, patient stratification, and treatment outcome prediction. This approach paves the way for biomarker-guided classification in clinical practice, ultimately leading to more tailored therapeutic interventions.

There were several limitations in this study. First, as a single-center retrospective study, the findings may be limited in generalizability, and larger, multi-center studies are needed for validation. Second, most of the enrolled study patients and controls were women, and there might be sex-related differences. Third, information on IC duration was not available in the clinical records of the present study. Fourth, there might be intra-individual variations. Currently, there is no consensus or standardized guideline regarding the optimal timing for urine sample collection for the measurement of biomarkers, which may have introduced variability in the results. Moreover, systemic inflammatory diseases, comorbidities, and local bladder insults could have influenced urinary biomarker expression, adding complexity to the interpretation of findings. Additionally, although the controls were carefully selected to minimize bladder-related pathology, stress urinary incontinence is still a form of lower urinary tract disease and may not fully represent a truly normal bladder without underlying dysfunction. In the analysis of treatment outcomes, although most patients across all clusters underwent intravesical PRP injections and the treatment distribution was similar, this variability may still have influenced the results. Finally, we selected K-means clustering due to its interpretability, simplicity, and suitability for exploratory analyses in studies with relatively small to moderate sample sizes. Further studies incorporating more advanced machine learning or deep learning methods are warranted to validate and expand upon our findings.

## 4. Materials and Methods

### 4.1. Patients

We retrospectively analyzed urinary biomarker data from a cohort of 127 IC/BPS patients collected between February 2014 and December 2018 at the Department of Urology, Hualien Tzu Chi Hospital. Enrolled IC/BPS patients underwent cystoscopic hydrodistention and were diagnosed according to the proposed guidelines of ESSIC, which constituted “chronic pelvic pain, pressure, or discomfort perceived to be related to the urinary bladder accompanied by at least one other urinary symptom, such as persistent urge to void or urinary frequency, for more than 6 months” and the exclusion of potentially similar diseases [12].

Enrolled patients underwent cystoscopy with hydrodistention under general anesthesia, and they were classified as ESSIC type 1 or 2 (without and with glomerulations, respectively). Patients with Hunner’s lesion (ESSIC type 3) were excluded from this study. Other exclusion criteria for analysis included active urinary tract infection, neurogenic voiding dysfunction (including cerebrovascular accident, spinal cord injury, multiple sclerosis, and Parkinson’s disease), a history of urinary tract malignancy or tuberculosis, a history of bladder surgery/or traumatic injury, a history of urethral or prostate surgery, a history of pelvic radiation, a history of nephrotic or nephritic syndrome, urolithiasis, and/or impaired renal function (serum creatinine > 2.0 mg/dL).

Additionally, 30 women with genuine stress urinary incontinence who were ready to undergo anti-incontinence sling surgeries served as controls. All controls did not have other significant lower urinary tract symptoms (defined as International Prostate Symptom Score < 6) or other proven storage or voiding dysfunction in video-urodynamic studies.

### 4.2. Clinical Investigation

The clinical assessment for enrolled IC/BPS patients included the ICSI, ICPI, OSS, and VAS pain score, all of which were performed by the same experienced clinical investigator to ensure consistency and minimize inter-rater variability. We also recorded the findings on cystoscopic hydrodistention, including MBC under general anesthesia, and the grade of glomerulations. The grading of glomerulations was determined by the severity observed during the examination of five bladder regions (anterior, posterior, left lateral, right lateral, and bottom) [22]. The grades were defined as follows: normal (0), petechiae present in at least two quadrants (I), large areas of submucosal bleeding (II), diffuse and widespread submucosal bleeding (III), and mucosal disruption with or without associated bleeding (IV).

The bladder treatments for enrolled IC/BPS patients in this study included intravesical hyaluronic acid installation, intravesical platelet-rich plasma injection, and intravesical botulinum toxin injection. The clinical indications for the selected treatments were based on patients with persistent symptoms despite undergoing behavioral therapy and pharmacotherapy. All patients had received cystoscopic hydrodistention prior to these interventions. The overall treatment outcomes were determined via telephone interviews at 3 months after treatment. Treatment response was evaluated using GRA, a 7-point scale ranging from −3 to +3, where +3 denotes markedly better, +2 denotes moderately better, +1 denotes mildly better, 0 denotes unchanged, −1 denotes mildly worse, −2 denotes moderately worse, and −3 denotes markedly worse [23]. A GRA score of +2 or +3 was defined as a successful outcome [24].

### 4.3. Urinary Biomarkers Investigation

Urine samples were collected from all enrolled study patients, and they were self-voided when the subjects reported a full bladder sensation. We performed urinalysis simultaneously to confirm an infection-free status before urine samples were stored. In total, 50 mL of urine was placed on ice immediately and transferred to the laboratory for preparation. The samples were centrifuged at 1800 rpm for 10 min at 4 °C. The supernatants were separated into aliquots in 1.5 mL tubes (1 mL per tube) and stored at −80 °C. Before further analyses were performed, the frozen urine samples were centrifuged at 12,000 rpm for 20 min at 4 °C, and the supernatants were used for subsequent measurements.

### 4.4. Quantification of Urinary Inflammatory and Oxidative Stress Biomarkers

The targeted analytes in the urine specimen included 10 inflammatory proteins/cytokines and 3 oxidative stress biomarkers (8-OHdG, 8-isoprostane, and TAC). The 10 inflammatory proteins/cytokines were measured using human cytokine/chemokine magnetic bead-based panel kits (Millipore, Darmstadt, Germany). The used panel kits included catalogue number HCYTMAG-60K-PX30 (Eotaxin, IL-2, IL6, IL-8, CXCL10, MCP-1, MIP-1β, RANTES, and TNFα) and catalogue number HADK2MAG-61K (NGF). The detailed procedures were similar to the previous study [7].

The quantification of 8-OHdG, 8-isoprostane, and TAC in urine samples was performed in accordance with the respective manufacturer’s instructions (8-OHdG ELISA kit, Biovision, Waltham, MA, USA; 8-isoprostane ELIZA kit, Enzo, Farmingdale, NY, USA; Total Antioxidant Capacity Assay Kit, Abcam, Cambridge, MA, USA). The detailed procedures were similar to the previous study [8].

### 4.5. Statistical Analysis

Continuous variables were represented by means ± standard deviations, while categorical variables were represented by numbers and percentages. Mean differences in clinical data, as well as the levels of urinary biomarkers among groups, were analyzed using one-way analysis of variance, and a post hoc test was performed via Bonferroni’s correction. Linear regression analysis with Pearson correlation was carried out to determine the relationship between clinical characteristics and the levels of urinary biomarkers. All calculations were performed using SPSS Statistics for Windows, Version 20.0 (IBM Corp., Armonk, NY, USA). If the *p*-value is less than 0.05, the difference is considered statistically significant.

### 4.6. Cluster Analysis Model

K-means clustering is basically a partitioning method applied to analyze data and treats the observations of the data as objects based on locations and distances between various input data points. Partitioning the objects into mutually exclusive clusters (K) is carried out in such a fashion that objects within each cluster remain as close as possible to each other but as far as possible from objects in other clusters. Cluster analysis, a machine learning model, was performed using R version 3.2.5, which was available at the time of analysis. The R software is open-access and remains available at http://www.r-project.org (accessed on 10 April 2025). K-means, an unsupervised learning method, is used for the clustering of urinary biomarkers. The NbClust package (version 3.0.1) is used to determine the optimal number of clusters [25].

## 5. Conclusions

This study demonstrates that the K-means clustering of urinary biomarkers provides a novel approach to IC/BPS classification, revealing distinct biomarker-driven subgroups with differing clinical characteristics and treatment responses. These findings highlight the heterogeneity of IC/BPS and the potential role of biomarker-guided precision medicine in its management. Further validation in larger cohorts is needed to confirm these findings and enhance their clinical applicability.

## Figures and Tables

**Figure 1 ijms-26-03712-f001:**
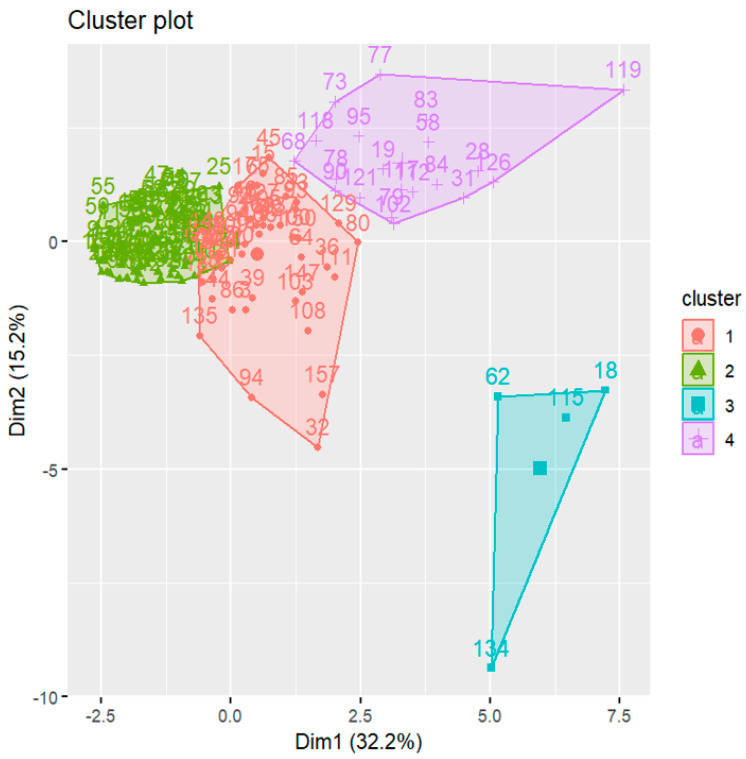
Cluster plot generated by K-means clustering (N = 157, including IC/BPS patients and controls), showing 4 clusters with sizes of 53, 80, 4, and 20, respectively.

**Figure 2 ijms-26-03712-f002:**
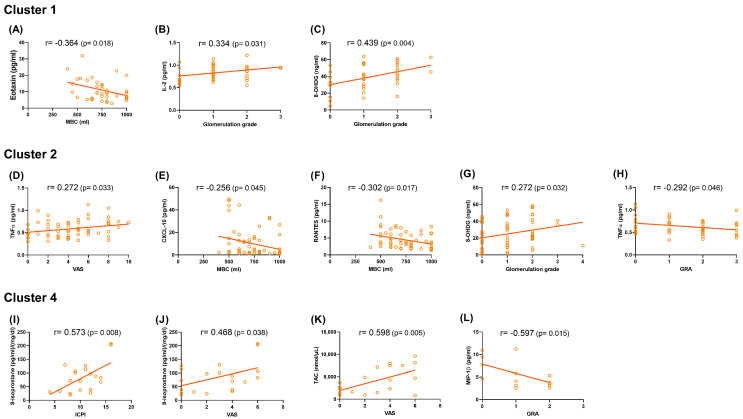
Correlation coefficients of urine cytokine levels and clinical characteristics within each cluster (excluding controls). (**A**–**C**) Cluster 1: Correlations between Eotaxin and MBC (**A**), IL-8 and glomerulation grade (**B**), and 8-OHdG and glomerulation grade (**C**). (**D**–**H**) Cluster 2: Correlations between TNF-α and VAS pain score (**D**), CXCL10 and MBC (**E**), RANTES and MBC (**F**), 8-OHdG and glomerulation grade (**G**), and TNF-α and GRA (**H**). (**I**–**L**) Cluster 4: Correlations between 8-isoprostane and ICPI (**I**), 8-isoprostane and VAS (**J**), TAC and VAS (**K**), and MIP-1β and GRA (**L**).

**Table 1 ijms-26-03712-t001:** Clinical characteristics of IC/BPS patients and controls.

	IC/BPS	Control	*p* Value
Number	127	30	
Age	54.6 ± 12.6	58.9 ± 10.8	0.083
Hypertension	29 (22.8%)	6 (20.0%)	0.737
Diabetes mellitus	16 (12.6%)	4 (13.3%)	1.000
Sex			0.075
Male	13 (10.2%)	0 (0.0%)	
Female	114 (89.8%)	30 (100.0%)	
ESSIC type			
Type I	37 (29.1%)		
Type II	90 (70.9%)		
Clinical characteristics			
ICSI	10.2 ± 4.5	NA	
ICPI	10.2 ± 4.0		
OSS	20.6 ± 7.9		
VAS	4.2 ± 2.7		
MBC (mL)	711.4 ± 179.5		
Glomerulation grade			
0	37 (29.1%)	NA	
1	41 (32.3%)		
2	43 (33.9%)		
3	5 (3.9%)		
4	1 (0.8%)		
Treatment			
Intravesical platelet-rich plasma injection	66 (68.8%)	NA	
Intravesical botulinum toxin injection	17 (17.7%)		
Intravesical hyaluronic acid installation	13 (13.5%)		
GRA		NA	
0	21 (21.9%)		
+1	28 (29.2%)		
+2	31 (32.3%)		
+3	16 (16.7%)		

IC/BPS, Interstitial cystitis/bladder pain syndrome; ESSIC, European Society for the Study of Interstitial Cystitis; ICSI, Interstitial Cystitis Symptom Index; ICPI, Interstitial Cystitis Problem Index; OSS, O’Leary–Sant symptom score; VAS, visual analogue scale pain score; MBC, maximal bladder capacity under anesthesia; GRA, Global Response Assessment; NA, not available.

**Table 2 ijms-26-03712-t002:** Urine biomarker profiles between IC/BPS patients and controls.

	IC/BPS	Control	*p* Value
Number	127	30	
Urine biomarkers *			
Eotaxin	9.46 ± 8.68	5.51 ± 4.32	0.017
IL-2	0.76 ± 0.18	0.84 ± 0.23	0.032
IL-6	2.52 ± 4.84	1.90 ± 3.50	0.509
IL-8	15.70 ± 22.59	24.87 ± 57.57	0.163
CXCL10	57.52 ± 116.84	50.69 ± 192.93	0.803
MCP-1	358.76 ± 410.18	172.38 ± 131.90	0.015
MIP-1β	3.48 ± 3.36	2.77 ± 2.16	0.269
RANTES	10.02 ± 8.38	7.44 ± 8.01	0.129
TNFα	0.82 ± 0.61	0.93 ± 0.63	0.405
NGF	0.40 ± 0.27	0.26 ± 0.08	0.006
8-OHdG	34.66 ± 18.53	18.89 ± 13.84	<0.001
8-isoprostane	42.49 ± 36.48	19.66 ± 19.29	0.001
TAC	1975.72 ± 1807.06	1207.35 ± 1037.81	0.027

* Units: All pg/mL, except ng/mL in 8-OHdG, and nmol/μL in TAC. Data are presented as mean ± standard deviation. IC/BPS, Interstitial cystitis/bladder pain syndrome; IL, interleukin; CXCL10, chemokine (C-X-C motif) ligand 10; MCP-1, macrophage chemoattractant protein-1; MIP-1β, macrophage inflammatory protein 1β; RANTES, regulated upon activation, normal T cell expressed, and presumably secreted; TNFα, tumor necrosis factor α; NGF, nerve growth factor; 8-OHdG, 8-hydroxy-2-deoxyguanosine; TAC, total antioxidant capacity.

**Table 3 ijms-26-03712-t003:** Demographic and clinical characteristics of IC/BPS patients and controls categorized by K-means clustering of urine biomarker profiles.

	Cluster					*p* Value
	Cluster 1	Cluster 2	Cluster 3	Cluster 4	Total	
Number	53 (33.8%)	80 (51.0%)	4 (2.5%)	20 (12.7%)	157 (100%)	
Age	55.68 ± 12.43	55.80 ± 12.48	64.00 ± 9.20	51.60 ± 11.39	55.43 ± 12.31	0.267
Sex						0.047
Male	2 (3.8%)	6 (7.5%)	0 (0.0%)	5 (25.0%)	13 (8.3%)	
Female	51 (96.2%)	74 (92.5%)	4 (100.0%)	15 (75.0%)	144 (91.7%)	
Type						0.054
ESSIC type I IC/BPS	10 (27.0%)	23 (62.2%)	1 (2.7%)	3 (8.1%)	37 (100%)	
ESSIC type II IC/BPS	32 (35.6%)	39 (43.3%)	2 (2.2%)	17 (18.9%)	90 (100.0)	
Control	11 (36.7%)	18 (60.0%)	1 (3.3%)	0 (0.0%)	30 (100.0%)	
Clinical characteristics *						
ICSI	10.2 ± 4.8	10.4 ± 4.5	6.7 ± 3.1	10.3 ± 3.9	10.2 ± 4.5	0.575
ICPI	9.8 ± 4.4	10.5 ± 3.9	7.3 ± 2.1	10.5 ± 3.2	10.2 ± 4.0	0.471
OSS	20.2 ± 8.6	20.9 ± 7.8	14.0 ± 4.6	21.2 ± 6.6	20.6 ± 7.9	0.495
VAS	4.9 ± 2.5	4.2 ± 2.8	5.3 ± 1.2	2.8 ± 2.3	4.2 ± 2.7	0.033
MBC (mL)	747.1 ± 169.9	731.5 ± 172.1	666.7 ± 57.7	581.0 ± 183.7	711.4 ± 179.5	0.003
Glomerulation grade						0.072
0	10 (23.8%)	23 (37.1%)	1 (33.3%)	3 (15.0%)	37 (29.1%)	
1	18 (42.9%)	18 (29.0%)	2 (66.7%)	3 (15.0%)	41 (32.3%)	
2	12 (28.6%)	18 (29.0%)	0 (0.0%)	13 (65.0%)	43 (33.9%)	
3	2 (4.8%)	2 (3.2%)	0 (0.0%)	1 (5.0%)	5 (3.9%)	
4	0 (0.0%)	1 (1.6%)	0 (0.0%)	0 (0.0%)	1 (0.8%)	
Treatment *						0.326
Intravesical platelet-rich plasma injection	21 (70.0%)	30 (63.8%)	1 (33.3%)	14 (87.5%)	66 (68.8%)	
Intravesical botulinum toxin injection	4 (13.3%)	11 (23.4%)	1 (33.3%)	1 (6.3%)	17 (17.7%)	
Intravesical hyaluronic acid installation	5 (16.7%)	6 (12.8%)	1 (33.3%)	1 (6.3%)	13 (13.5%)	
GRA *						0.348
0	6 (20.0%)	11 (23.4%)	0 (0.0%)	4 (25.0%)	21 (21.9%)	
+1	11 (36.7%)	12 (25.5%)	0 (0.0%)	5 (31.3%)	28 (29.2%)	
+2	8 (26.7%)	15 (31.9%)	1 (33.3%)	7 (43.8%)	31 (32.3%)	
+3	5 (16.7%)	9 (19.1%)	2 (66.7%)	0 (0.0%)	16 (16.7%)	
GRA ≧ +2 (%)	13 (43.3%)	24 (51.1%)	3 (100.0%)	7 (43.8%)	47 (49.0%)	0.353
GRA = +3 (%)	5 (16.7%)	9 (19.1%)	2 (66.7%)	0 (0.0%)	16 (16.7%)	0.036 *

* Excluding the data from controls. IC/BPS, interstitial cystitis/bladder pain syndrome; ESSIC, European Society for the Study of Interstitial Cystitis; ICSI, Interstitial Cystitis Symptom Index; ICPI, Interstitial Cystitis Problem Index; OSS, O’Leary–Sant symptom score; VAS, visual analogue scale pain score; MBC, maximal bladder capacity under anesthesia; GRA, Global Response Assessment.

**Table 4 ijms-26-03712-t004:** Urine biomarker profiles of IC/BPS patients and controls categorized by K-means clustering.

Urine Biomarkers *	Cluster					*p* Value	Post Hoc
	Cluster 1 (n = 53, 33.8%)	Cluster 2 (n = 80, 51.0%)	Cluster 3 (n = 4, 2.5%)	Cluster 4 (n = 20, 12.7%)	Total (N = 157, 100%)		
Eotaxin	10.66 ± 6.06	3.94 ± 2.59	21.26 ± 18.97	20.09 ± 9.36	8.71 ± 8.17	<0.001	2 < 1 < 3, 4
IL-2	0.87 ± 0.20	0.69 ± 0.15	0.97 ± 0.17	0.85 ± 0.16	0.78 ± 0.19	<0.001	2 < 1, 3, 4
IL-6	2.34 ± 2.94	1.06 ± 1.02	17.33 ± 16.89	4.95 ± 6.34	2.40 ± 4.61	<0.001	1, 2, 4 < 3 & 2 < 4
IL-8	24.02 ± 22.45	6.56 ± 7.82	156.54 ± 112.50	15.84 ± 14.89	17.45 ± 32.27	<0.001	1, 2, 4 < 3 & 2 < 1
CXCL10	55.40 ± 41.70	8.85 ± 12.07	694.11 ± 480.81	120.23 ± 78.23	56.22 ± 133.99	<0.001	2 < 1 < 4 < 3
MCP-1	302.47 ± 158.07	145.63 ± 97.77	529.10 ± 362.76	1046.79 ± 603.13	323.14 ± 380.17	<0.001	2 < 1, 3 < 4
MIP-1β	4.03 ± 2.55	1.98 ± 1.23	13.29 ± 10.58	5.02 ± 2.79	3.34 ± 3.17	<0.001	2 < 1, 4 < 3
RANTES	12.11 ± 6.24	4.34 ± 2.62	11.66 ± 6.59	22.97 ± 10.34	9.52 ± 8.34	<0.001	2 < 1 < 4 & 3 < 4
TNFα	1.12 ± 0.88	0.61 ± 0.19	1.98 ± 0.41	0.82 ± 0.27	0.84 ± 0.61	<0.001	2 < 1 < 3 & 4 < 3
NGF	0.37 ± 0.22	0.40 ± 0.31	0.32 ± 0.11	0.31 ± 0.07	0.38 ± 0.25	0.512	
8-OHdG	36.41 ± 15.07	22.78 ± 16.38	28.90 ± 18.83	55.04 ± 11.11	31.65 ± 18.76	<0.001	2 < 1< 4 & 3 < 4
8-isoprostane	33.70 ± 23.90	28.28 ± 23.46	64.22 ± 50.83	84.04 ± 54.55	38.13 ± 35.00	<0.001	1, 2 < 4
TAC	1758.7 ± 1238.8	1290.0 ± 977.2	2432.8 ± 2199.1	4049.8 ± 2935.1	1828.9 ± 1711.6	<0.001	1, 2 < 4

* Units: All pg/mL, except ng/mL in 8-OHdG, and nmol/μL in TAC. Data are presented as mean ± standard deviation. IC/BPS, Interstitial cystitis/bladder pain syndrome; IL, interleukin; CXCL10, chemokine (C-X-C motif) ligand 10; MCP-1, macrophage chemoattractant protein-1; MIP-1β, macrophage inflammatory protein 1β; RANTES, regulated upon activation, normal T cell expressed, and presumably secreted; TNFα, tumor necrosis factor α; NGF, nerve growth factor; 8-OHdG, 8-hydroxy-2-deoxyguanosine; TAC, total antioxidant capacity.

**Table 5 ijms-26-03712-t005:** Summarized characteristics of each cluster.

	Cluster 1	Cluster 2	Cluster 3	Cluster 4
**Composition**	53 individuals (33.8% of cohort) Including ESSIC type 1 and 2 IC/BPS patients (18.9% and 60.4%) and control (20.8%)	80 individuals (51.0% of cohort) Including most ESSIC type 1 IC/BPS patients (n = 23, 62.2%) and controls (n = 18, 60%)	4 individuals (2.5% of cohort)	20 individuals (12.7% of cohort) A high proportion of ESSIC type 2 IC/BPS patients (n = 17, 85%)
**Clinical Characteristics ***	Higher VAS pain score, higher MBC	Higher VAS pain score, higher MBC	Not discussed	Lower VAS pain score, lower MBC
**Urine Biomarker Profiles**	Intermediate biomarker levels (higher than Cluster 2 but lower than Cluster4): Eotaxin, IL-2, IL-8, CXCL10, RANTES, and 8-OHdG	Lowest levels of most biomarkers among all clusters	Considered outliers with unique biomarker expressions	Highest levels of oxidative stress biomarkers (8-OHdG, 8-isoprostane, and TAC) and inflammatory biomarkers (Eotaxin, IL-6, CXCL10, MCP-1, and RANTES)
**T** **reatment Outcomes (GRA) ***	43.3% of patients achieving moderate to marked improvement (GRA ≧ +2)	51.1% of patients achieving moderate to marked improvement (GRA ≧ +2)	Not discussed	0% of patients achieving marked improvement (GRA = +3) (significantly lower compared to other clusters)
**Clinical Correlations ***	Weak to moderate correlations between urine biomarkers and cystoscopic hydrodistention parameters (MBC, glomerulation grade)	Weak correlations between urine biomarkers and various clinical parameters, including pain severity, cystoscopic hydrodistention parameters (MBC and glomerulation grade), and treatment response (GRA)	Not discussed	Moderate to strong correlations between urine biomarkers and clinical symptoms, pain severity, and treatment response (GRA)

* Excluding the data from controls. IC/BPS, Interstitial cystitis/bladder pain syndrome; ESSIC, European Society for the Study of Interstitial Cystitis; VAS, visual analogue scale pain score; MBC, maximal bladder capacity under anesthesia; GRA, Global Response Assessment; IL, interleukin; CXCL10, chemokine (C-X-C motif) ligand 10; MCP-1, macrophage chemoattractant protein-1; RANTES, regulated upon activation, normal T cell expressed, and presumably secreted; 8-OHdG, 8-hydroxy-2-deoxyguanosine; TAC, total antioxidant capacity.

## Data Availability

The original contributions presented in this study are included in the article/Appendix A. Further inquiries can be directed to the corresponding author.

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
