# Peer review of "Applying K-Means Cluster Analysis to Urinary Biomarkers in Interstitial Cystitis/Bladder Pain Syndrome: A New Perspective on Disease Classification"

_ijms, 2025, doi:10.3390/ijms26083712_

Round 1
Reviewer 1 Report
Comments and Suggestions for Authors
The study employs K-means cluster analysis to investigate urinary biomarker profiles in IC/BPS patients, offering a novel approach to disease classification by identifying subtypes with distinct clinical and biochemical characteristics. Some issues need to be addressed.
1) The ethical protocol requires more detailed explanation in the 'Materials and Methods' section, explicitly documenting institutional review board approval and informed consent procedures;
2) The rationale for excluding ESSIC type 3 patients needs clear justification;
3) The control group's symptom assessment is problematic, as the International Prostate Symptom Score is inappropriate for an all-female population;
4) Clinical investigations should implement a multi-rater approach, involving at least two independent clinicians to reduce subjective assessment bias;
5) The treatment outcome assessment method, limited to a 3-month telephone interview, requires additional context about potential long-term follow-up;
6) The definition of treatment success (GRA score of +2 or +3) needs supporting scientific references;
7) Authors should provide a comprehensive explanation for selecting specific oxidative stress biomarkers (8-OHdG, 8-isoprostane, and TAC) from the available options;
8) The protein testing methodology requires further scrutiny, particularly the testing kit's validation and the reasons behind the insignificant NGF differences across clusters;
9) The correlation analysis method should be explicitly detailed in the statistical analysis section;
10) The justification for choosing K-Means clustering over alternative deep learning methods needs clarification, ideally with a comparative analysis;
11) A comparative cluster analysis should be presented, both with and without control group inclusion;
12) Table 1 should be expanded to include additional patient characteristics such as IC duration and comorbidities;
13) The control group composition should be diversified by including male participants to improve demographic representation;
14) The unusually large Maximum Bladder Capacity of 711.4±179.5 in IC patients requires detailed discussion;
15) Clear clinical indications must be provided for the selected treatments (intravesical platelet-rich plasma, botulinum toxin injection, and hyaluronic acid installation);
16) A detailed supplementary file characterizing Cluster 3 should be developed;
17) The proportion of ESSIC type 1 and 2 patients in Cluster 1 needs to be explicitly reported;
18) The apparent contradiction between VAS pain scores and Maximum Bladder Capacity requires a comprehensive scientific explanation, exploring potential underlying mechanisms that might account for this unexpected relationship.
Author Response
Reviewer 1
The study employs K-means cluster analysis to investigate urinary biomarker profiles in IC/BPS patients, offering a novel approach to disease classification by identifying subtypes with distinct clinical and biochemical characteristics. Some issues need to be addressed.
- The ethical protocol requires more detailed explanation in the 'Materials and Methods' section, explicitly documenting institutional review board approval and informed consent procedures;
Reply: Thank you for your comment. The relevant ethical protocol details have been provided in the 'Institutional Review Board Statement' section, as required by the IJMS journal. (See P13, Lines 15–19)
- The rationale for excluding ESSIC type 3 patients needs clear justification;
Reply: Thank you for your comment.
- ESSIC type 3 IC/BPS patients (Hunner’s IC) typically exhibit significantly extremely higher levels of urinary cytokines compared to those with non-Hunner IC, as reported in previous studies. In addition, they account for only approximately 15% of the overall IC/BPS population, making their representation relatively limited.
- The identification of Hunner’s lesions (i.e., classification as ESSIC type 3) requires cystoscopic evaluation, which, while considered the gold standard. Clinically, Hunner-type IC (ESSIC type 3) also presents with different features and underlying pathophysiology compared to non-Hunner IC.
Based on these considerations—including differences in biomarker profiles, diagnostic approaches, and disease characteristics—we excluded ESSIC type 3 patients from the current study to maintain a more homogeneous study population and reduce potential confounding factors.
- The control group's symptom assessment is problematic, as the International Prostate Symptom Score is inappropriate for an all-female population;
Reply: Thank you for your comment. The IPSS was originally developed to assess male lower urinary tract dysfunction (LUTD); however, subsequent studies have demonstrated that it can also be applied to evaluate female LUTD and may provide an initial guide for managing voiding dysfunction in women (Int Urogynecol J. 2013 Feb;24(2):263).
In our study, the IPSS was used solely as a screening tool to identify suitable female participants for the control group (IPSS < 6), rather than for symptom evaluation or outcome analysis. Moreover, previous studies investigating urinary biomarkers in IC/BPS have adopted similar criteria for selecting control subjects (ref. 7, ref. 8).
- Clinical investigations should implement a multi-rater approach, involving at least two independent clinicians to reduce subjective assessment bias;
Reply: Thank you for your comment. We appreciate the reviewer’s valuable suggestion. We agree that involving multiple independent clinicians can help reduce subjective assessment bias in clinical investigations. In our study, all symptom assessments were performed by the same experienced clinical investigator, which helped to ensure consistency and minimize inter-rater variability (We added this description in the text. (P10 Line 29-32).
While a multi-rater approach would certainly enhance methodological rigor, it is not yet a commonly adopted practice in similar studies in this field.
- The treatment outcome assessment method, limited to a 3-month telephone interview, requires additional context about potential long-term follow-up;
Reply: Thank you for your comment. A 3-month follow-up is a commonly adopted time point for evaluating treatment outcomes in IC/BPS, particularly after interventions such as hyaluronic acid instillation or intravesical injection therapy. This period is generally recognized as the optimal window to observe peak therapeutic response. While longer-term follow-up (e.g., 6 months) may provide additional information, treatment effects often diminish over time, which could make it more difficult to accurately assess the true efficacy of the intervention.
- The definition of treatment success (GRA score of +2 or +3) needs supporting scientific references;
Reply: Thank you for your comment. We have added the supporting scientific reference. (P11, Line 14-15, ref. 24)
- Authors should provide a comprehensive explanation for selecting specific oxidative stress biomarkers (8-OHdG, 8-isoprostane, and TAC) from the available options;
Reply: Thank you for your comment. Previous studies have demonstrated that IC/BPS patients exhibit distinct urinary oxidative stress biomarker and inflammatory cytokine profiles compared to controls (ref. 8). (P3, Line 15-17)
In that study, 8-OHdG, 8-isoprostane, and TAC were specifically investigated and reported as key oxidative stress biomarkers (ref. 8). Therefore, we selected the same set of biomarkers in the present study to ensure consistency and to build upon established findings.
- The protein testing methodology requires further scrutiny, particularly the testing kit's validation and the reasons behind the insignificant NGF differences across clusters;
Reply: Thank you for your comment.
- The protein testing kits used in this study have been widely applied in previously published IC/BPS studies by different research groups (including ref. 7, ref. 8, and ref. 20). The validity of these kits can be considered acceptable based on current literature.
- In our study, urinary NGF levels were significantly higher in IC/BPS patients compared to controls (Table 2), which is consistent with previous findings. However, no significant differences in NGF levels were observed across the identified clusters (Table 4). This suggests that while urinary NGF is useful for distinguishing IC/BPS patients from controls, it may not be sufficient to differentiate subtypes (different Clusters) within the IC/BPS population. The clustering was based on the combined profile of 13 urinary biomarkers, not NGF alone. This also highlights the importance of a comprehensive assessment using the overall biomarker profile—such as through K-means cluster analysis—rather than relying on individual biomarkers in isolation.
We have added above descriptions in the text. (P8, Line 15-24)
- The correlation analysis method should be explicitly detailed in the statistical analysis section;
Reply: Thank you for your comment. We have added the description of correlation analysis method in the statistical analysis section:
Linear regression analysis with Pearson correlation was carried out to determine the relationship between clinical characteristics and the levels of urine biomarkers. (P12, Line 14-16)
- The justification for choosing K-Means clustering over alternative deep learning methods needs clarification, ideally with a comparative analysis;
Reply: Thank you for your comment. This study represents one of the early efforts to apply K-Means clustering in the context of urinary biomarker-based research for IC/BPS. As such, there is currently a limited number of studies validating or comparing different clustering or deep learning approaches in this specific field. We chose K-Means for its interpretability, simplicity, and suitability for exploratory analysis, particularly in studies with relatively small to moderate sample sizes. We agree that further studies incorporating more advanced machine learning or deep learning methods are warranted to validate and build upon our findings. We have added this statement to the Limitations section of the manuscript.
“Finally, we selected K-Means clustering due to its interpretability, simplicity, and suitability for exploratory analyses in studies with relatively small to moderate sample sizes. Further studies incorporating more advanced machine learning or deep learning methods are warranted to validate and expand upon our findings.” (P9, Line 30-34)
- A comparative cluster analysis should be presented, both with and without control group inclusion;
Reply: Thank you for your comment. Whether or not to include control subjects in cluster analysis depends on “the purpose of the analysis and how the results are interpreted”.
In this study, we included controls in the clustering process to reflect a real-world situation where patients have not yet been diagnosed with IC/BPS. This approach allows us to explore how urinary biomarker profiles form natural groupings across both IC/BPS patients and controls. One advantage is that it reflects a subset of IC/BPS patients whose urinary biomarker profiles are similar to those of controls. Including controls also makes future prospective validation easier.
For comparison, we also performed a separate cluster analysis excluding the controls (We have added this cluster analysis to “Supplementary Figure 2”). This represents a scenario where all patients are confirmed IC/BPS cases, and the goal is to identify different subtypes within the disease group. As shown in Supplementary Figure 2 (figure below), this analysis also produced clear clusters, including one group that appeared to be outliers (Cluster 4 in Supplementary figure 2).
We have added the below sentences in the section of Results:
“To further evaluate the clustering performance within confirmed IC/BPS patients, an additional cluster analysis excluding the control group was performed (Supplementary Figure 2). This supplementary analysis yielded four distinct clusters, including one potential outlier group, and supports the robustness of the urinary biomarker-based clustering approach.” (P6, Line 15-19)
- Table 1 should be expanded to include additional patient characteristics such as IC duration and comorbidities;
Reply: Thank you for your comment. Information on IC duration was not available in the clinical records at the time of this study, and we have acknowledged this as a limitation in the revised manuscript (P9, Line 17-18). Comorbidity data, including hypertension and diabetes mellitus (DM), have been added to Table 1 as requested.
- The control group composition should be diversified by including male participants to improve demographic representation;
Reply: Thank you for your comment. Various factors, including sex, may influence urinary biomarker expression; however, current evidence remains limited and inconclusive. While sex-related differences in biomarker levels are possible, IC/BPS predominantly affects women, with approximately 80-85% of patients being female. Accordingly, many previous IC/BPS biomarker studies have also used female participants as controls (ref. 7, ref. 8, ref. 16).
To address this concern, we re-analyzed the data after excluding male participants, and the results remained consistent with those of the entire cohort (the p-values in Table 2 remained significant). We have also acknowledged the potential influence of sex on urinary biomarker expression in the Limitations section of the manuscript. (P9, Line 15-17)
- The unusually large Maximum Bladder Capacity of 711.4±179.5 in IC patients requires detailed discussion;
Reply: Thank you for your comment. In our study, IC/BPS patients were enrolled based on the ESSIC criteria, which rely primarily on clinical symptoms, rather than the stricter NIDDK criteria commonly used in earlier research studies. As a result, the included patients may represent a broader spectrum of IC/BPS severity, including those with relatively larger bladder capacities. Notably, our findings are consistent with previous studies that also used the ESSIC criteria for patient inclusion (ref. 7, ref. 16).
- Clear clinical indications must be provided for the selected treatments (intravesical platelet-rich plasma, botulinum toxin injection, and hyaluronic acid installation);
Reply: Thank you for your comment. “The clinical indications for the selected treatments were based on patients with persistent symptoms despite undergoing behavioral therapy and pharmacotherapy. All patients had received cystoscopic hydrodistention prior to these interventions.”
We have added this description to the revised manuscript. (P11, Line 6-9)
- A detailed supplementary file characterizing Cluster 3 should be developed;
Reply: Thank you for your comment. The demographic and clinical characteristics, as well as the urinary biomarker profiles of patients in Cluster 3, are already presented in Table 3 and Table 4, respectively. We would be glad to provide a supplementary file if the reviewer could specify the additional information required.
- The proportion of ESSIC type 1 and 2 patients in Cluster 1 needs to be explicitly reported;
Reply: Thank you for your comment. We have added the detailed proportions of ESSIC type 1 and type 2 IC/BPS patients in Cluster 1 in Table 5. Specifically, Cluster 1 is composed of 18.9% ESSIC type 1 patients, 60.4% ESSIC type 2 patients, and 20.8% control subjects.
18) The apparent contradiction between VAS pain scores and Maximum Bladder Capacity requires a comprehensive scientific explanation, exploring potential underlying mechanisms that might account for this unexpected relationship.
Reply: Thank you for your comment. We indeed found that “Cluster 4 patients had elevated oxidative stress markers (8-OHdG, 8-isoprostane, TAC) and inflammatory cytokines (Eotaxin, IL-6, CXCL10, MCP-1, and RANTES), along with lower maximum bladder capacity (MBC) and lower VAS pain scores.”
We proposed several possible mechanisms for this contradictory finding in the Discussion section as follows:
“These findings suggest that Cluster 4 patients experience pronounced oxidative stress and more severe bladder inflammation, manifesting as reduced bladder capacity yet lower VAS pain scores. One possible explanation for this contradictory finding (i.e., lower MBC yet lower VAS) is that chronic or intense inflammation can lead to partial nerve desensitization, tissue remodeling or fibrosis, altered pain thresholds, or behavioral adaptations (e.g., frequent voiding that minimizes overdistension), thereby diminishing perceived pain. Consequently, despite more severe structural damage and elevated oxidative stress and inflammatory markers, such alterations in sensory signaling and pain modulation might attenuate pain perception.”
This paragraph has been added to the Discussion section of the revised manuscript. (P7, Line 26-36)

Reviewer 2 Report
Comments and Suggestions for Authors
Dear authors,
Thank you for your valuable manuscript. This is a really clever point of view trying clustering IC/BPS subtypes. However, there are several issues to be discussed before a possible publication
- the control group consists of women with SUI, preparing for a continence surgery. I am afraid this is not a safe and totally independent group of patients, as SUI is still a LUTD, probably without urodynamic disturbances, but how sure could we feel about histopathological urothelium damage? I think that this could be discussed more extensively as a limitation.
- there is no control group for male population.
- I am not sure, if this model is appropriate for a non-bladder centric disease. Please, consider to exclude patients with co-morbidities with symptoms like IC/BPS.
- different treatment models could affect biomarkers physical history in different ways. You should discuss it extensively, as it could be a major factor fro non-comparable results.
Author Response
Reviewer 2
Dear authors,
Thank you for your valuable manuscript. This is a really clever point of view trying clustering IC/BPS subtypes. However, there are several issues to be discussed before a possible publication
the control group consists of women with SUI, preparing for a continence surgery. I am afraid this is not a safe and totally independent group of patients, as SUI is still a LUTD, probably without urodynamic disturbances, but how sure could we feel about histopathological urothelium damage? I think that this could be discussed more extensively as a limitation.
Reply: Thank you for your comment. We agree that finding a completely “ideal” control group in studies of bladder urothelium or urinary biomarkers is challenging. In our study, we selected women with stress urinary incontinence (SUI) who did not have other significant lower urinary tract symptoms (defined as an International Prostate Symptom Score < 6) or any documented storage or voiding dysfunction on video-urodynamic studies. These criteria were applied to ensure that their condition was “primarily due to urethral sphincter incompetence, without underlying bladder pathophysiology”. In the absence of bladder dysfunction, the urothelium and urinary biomarkers of these SUI patients are considered to “more closely reflect a non-pathological control state”.
This selection strategy has also been adopted in previously published studies in LUTD, including:
Study of IC bladder: BJU Int. 2013 Dec;112(8):1156
Study of DU bladder: J Urol. 2017 Jan;197(1):197
Study of IC urine biomarker: ref. 7, ref.8, ref. 16
We acknowledge the reviewer’s concern and have added this point as a limitation in the revised manuscript.
“Additionally, although the controls were carefully selected to minimize bladder-related pathology, stress urinary incontinence is still a form of lower urinary tract disease and may not fully represent a truly normal bladder without underlying dysfunction.” (P9, Line 24-27)
there is no control group for male population.
Reply: Thank you for your comment. Various factors, including sex, may influence urinary biomarker expression; however, current evidence remains limited and inconclusive. While sex-related differences in biomarker levels are possible, IC/BPS predominantly affects women, with approximately 80-85% of patients being female. Accordingly, many previous IC/BPS biomarker studies have also used female participants as controls (ref. 7, ref. 8, ref. 16).
To address this concern, we re-analyzed the data after excluding male participants, and the results remained consistent with those of the entire cohort (the p-values in Table 2 remained significant). We have also acknowledged the potential influence of sex on urinary biomarker expression in the Limitations section of the manuscript.(P9, Line 15-17)
I am not sure, if this model is appropriate for a non-bladder centric disease. Please, consider to exclude patients with co-morbidities with symptoms like IC/BPS.
Reply: Thank you for your comment. Enrolled IC/ BPS patients were diagnosed according to ESSIC guidelines, with the exclusion of potentially similar diseases. We have mentioned it in the section of Materials and methods. (P10 Line 3-9)
different treatment models could affect biomarkers physical history in different ways. You should discuss it extensively, as it could be a major factor fro non-comparable results.
Reply: Thank you for your comment. We have mentioned this limitation in the section of Limitation:
“In the analysis of treatment outcomes, although most patients across all clusters underwent intravesical PRP injections and the treatment distribution was similar, this variability may still have influenced the results.” (P9, Line 27-30)

Round 2
Reviewer 1 Report
Comments and Suggestions for Authors
Thank you to the authors for their detailed response. However, there is one issue that still needs to be addressed.
In Reference 24, it is stated that "the result was considered successful when patients reported an improvement of one or more points on the GRA scale." However, in the current study, the authors define treatment success as a GRA score of +2 or +3. How do the authors justify this discrepancy?
Author Response
Reviewer 1
In Reference 24, it is stated that "the result was considered successful when patients reported an improvement of one or more points on the GRA scale." However, in the current study, the authors define treatment success as a GRA score of +2 or +3. How do the authors justify this discrepancy?
Reply: Thank you for the comment. The citation of Reference 24 in the previous version was incorrect. We have now updated the reference accordingly. The revised citation (Pain Physician. 2012 May-Jun;15(3):197) corresponds to a study in which a GRA score ≥ +2 was used to define a successful outcome, which is consistent with the criteria applied in our current study. This correction has been made in the revised manuscript.
Reviewer 2 Report
Comments and Suggestions for Authors
Accepted in the revised version.
Author Response
Reviewer 2
Accepted in the revised version.
Reply: Thank you for the review and the comments.